# Palliative Care in Patients with End-Stage Renal Disease: A Meta Synthesis

**DOI:** 10.3390/ijerph182010651

**Published:** 2021-10-11

**Authors:** Nur Fithriyanti Imamah, Hung-Ru Lin

**Affiliations:** 1School of Nursing, College of Nursing, National Taipei University of Nursing and Health Sciences, Taipei 112303, Taiwan; fithriimamah@gmail.com; 2Department of Nursing, Faculty of Nursing, Universitas Muhammadiyah Kalimantan Timur, Samarinda 75124, Indonesia

**Keywords:** palliative care, end-stage renal disease, qualitative synthesis

## Abstract

End-stage renal disease is the last stage of chronic kidney disease and is associated with a decreased quality of life and life expectancy. This study aimed to explore palliative care with end-stage renal disease. Qualitative meta-synthesis was used as the study design. The search was performed for qualitative studies published until June 2021 and uses reciprocal translation and synthesis of in vivo and imported concepts. Five themes were included: Struggling to face the disease, experiencing deterioration, overcoming the challenges of dialysis, leading to a positive outlook, and preparing for the end of life. In facing chronic disease with life-limiting potential, patients experienced some negative feelings and deterioration in their quality of life. Adaptation to the disease then leads patients to a better outlook through increased spirituality and social status. Furthermore, by accepting the present condition, they started to prepare for the future. Increasing awareness of mortality leads them to discuss advance care (ACP) planning with healthcare professionals and families.

## 1. Introduction

End-stage renal disease (ESRD) is the last stage (5th stage) of Chronic Kidney Disease (CKD), with only 15% or less of the kidney functioning in filtration. A decrease in filtration is associated with a wide range of complications, such as hypertension, anaemia, malnutrition, bone disease, neuropathy, and decreased quality of life [1]. Treatment for end-stage renal disease is not only focused on dialysis or renal replacement therapy but also involves fluid and dietary management, medication, and medical engagement for scheduling treatment [2]. Furthermore, caring for patients with a life-limiting illness focuses on several aspects, including comfort, peace, and dignity. The transition from life to death is the greatest challenge in caring for patients with life-limiting illnesses. The caring process is influenced by culture, religion, and society [3].

The illness trajectory for patients with ESRD follows the illness trajectory of solid organ failure or might be similar to that of congestive heart failure and chronic obstructive lung disease, but it will be different in patients who choose not to undergo dialysis from the beginning [4]. Dialysis patients often experience decreases in functional status and quality of life, feel less independent and are unable to participate in social activities. Withdrawal from dialysis is associated with increased mortality in ESRD patients who are dialysis-dependent [5]. Most of the patients stop dialysis prior to death, while some older patients might choose to forgo dialysis due to their functional status and predicted poor outcomes [6,7,8,9,10]. A systematic review study reported that the elderly with dialysis have a similar survival rate with those who choose conservative management with a year survival [11]. Once the patient chooses to withdraw from dialysis, the nephrologist will coordinate with the palliative care team to prepare a peaceful and dignified death through comprehensive symptom control [5,9].

Providing palliative care to patients with ESRD includes areas of advance care planning, pain and symptom management, and bereavement support [5,12,13,14,15,16]. Most ESRD patients feel the importance of preparedness and plan ahead for death. Few patients start the end-of-life conversation at the beginning of dialysis. Most of them start during dialysis or when they are becoming ill [17]. When withdrawal from dialysis has been chosen, the nurses are responsible for the implementation of end-of-life care plans, including portable medical orders for life-sustaining treatment at the end of life, confirmation of the patient’s wish, site of death and its feasibility wish, recommendation of hospice if available, initiation of comprehensive symptom control (physical, emotional, and spiritual), discussion of the patient’s wishes for palliative sedation, anticipation of the development of uremic symptoms and preparation for treatment [5].

Globally, end-stage renal disease is included among the chronic diseases with life-limiting potential. There was a shift in kidney failure nursing studies, where qualitative studies increased compared to quantitative studies [17,18,19]. More studies preferred to investigate the personal experiences of patients living with chronic illnesses [20,21].

Studies of patients with end-stage renal disease mostly discuss the perception of illness and treatment, depression and mortality among patients, physical problems, and symptom management. While intervention programs for ESRD patients in this stage are important, first, it is necessary to explore the understanding of patients about their future life-limiting illness to influence their caregiving process and outcomes through published qualitative studies that explore the experience of patients with ESRD with palliative care.

Qualitative studies of this chronic illness have been conducted since 1960–1980 and began to increase in kidney failure subjects after 1990 [20]. The evolution of qualitative studies has led [22] to a need for synthesis in qualitative studies. Qualitative meta-synthesis is a qualitative method that analyses the findings of studies to create a new interpretation to be examined in practice. Several qualitative meta-syntheses have been conducted in the general population, a chronic illness, a chronic illness in a specific population, and patients with kidney failure [18,20,23,24]. This study aimed to explore palliative care with ESRD through identification and synthesize qualitative evidence for better palliative care for ESRD patients.

## 2. Materials and Methods

### 2.1. Design

This study is a qualitative meta synthesis study to analyze the qualitative studies related to palliative care in end-stage renal disease patients. This study uses reciprocal translation and synthesis of in vivo and imported concepts. The reviewer engages in consistent comparison of the study and conceptual synthesis, or the reviewer may import concepts from other studies to incorporate in the results [25].

### 2.2. Search Methods

The inclusion criteria included a primary qualitative study published in English on the palliative care of patients with end-stage renal disease. Papers were excluded if they were quantitative or mixed methods studies; review reports; studies in a non-end-stage renal disease population exploring the experience of facing a life-limiting illness; studies concerning solely deliberate self-harm or non-suicidal self-injury; opinions; reports by healthcare organizations; summaries or discussions of editorials; notes; letters; conference extracts; dissertations; or case reports.

### 2.3. Search Strategy

This study explored published studies of patients’ experiences of life-limiting conditions of end-stage renal disease identified by searching the following databases: Cumulative Index of Nursing and Allied Health Literature (CINAHL), MEDLINE, EBSCOHost, PubMed, and APA PsycArticles. The search was performed in June 2021 to identify papers published between January 1990 and June 2021. Three clusters of keywords were included (i) those that concern the topic of interest (end-stage renal disease, stage 5 renal disease, end-stage kidney disease, etc.), (ii) those that concern the participants (life-limiting, palliative care, etc.), and (iii) those that concern qualitative research (qualitative research, interviews, focus groups, content analysis, etc.). The final algorithm used (in the PubMed Web search tool) is provided in Table 1. Databases were searched individually using the cluster search terms, after which the results were combined, and duplicates were removed.

### 2.4. Search Outcomes

The documentation of the strategy has been included in the search strategy and study selection, quality assessment, and data analysis sections. Individual data analysis was conducted and then peer review was conducted to identify the included articles and validate dependability. A final theme was reached after cross-examination between the two authors. For clarity, the selection process is presented in a flowchart (Figure 1). Each reviewed reference and the decision process were retained for audit trials.

### 2.5. Quality Appraisal

In consideration of quality assurance, the included papers were examined retrospectively at the request of the journal using a tool that combined elements of STARLITE [26], RAMESES Checklist [27], and CASP instruments [28,29] by the 2 authors. The Critical Appraisal Skills Program (CASP), which is the most frequently used instrument [30], addresses all the principles and assumptions underpinning qualitative research, including statement of purpose; appropriateness of methodology, design and recruitment; data collection procedure; relationship between researcher and participant; ethics; and rigor of data analysis, clarity of findings and value of the research [31]. It is one of the instruments recommended by the Cochrane Collaboration [32] and has been used in several important thematic analyses of medical topics. A 3-point scale for each criterion (0 = criterion not met; 1/P = criterion partially met; 2/T = criterion totally met) was applied [33].

### 2.6. Data Abstraction

Screening was performed independently to search for full-text articles. The studies were assessed by 2 authors for the following characteristics: study purpose, study method, sampling and data collection technique, sample size and key characteristics of subjects, data analysis technique and rigor, and results. The results of the database searches were entered into a bibliographic software programme (EndNote® v.20 developed by Thomson ISI Researchsoft manufactured by Camelot UK Bidco Limited (Clarivate Analytics)) for automatic removal of duplicates. Then, all titles and abstracts and studies selected according to our inclusion criteria (defined earlier) were screened. If the abstract was not sufficient, the full text was read. Full texts of potentially relevant articles were examined and a second selection was performed. At this phase, each article’s reference list as a source for new articles has been overlooked.

### 2.7. Synthesis

Data analysis consisted of the discussion and conclusion sections of each included paper and a qualitative content analysis [34]. Related attributions and explanations were grouped together in sets, and then simple explanatory propositions were formulated. Data were then extracted from the studies into a table. The tabulation of the qualitative findings within a single matrix supported the fusion of narrative data. Member agreement was performed to achieve agreement about themes and sub-themes that were reviewed independently by adding or revising the items and then integrated into the final analysis.

## 3. Results 

A total of 3051 studies were retrieved through the database search. A total of 1234 studies were removed due to duplication and 1768 studies were rejected due to being mixed methods studies, review reports, non-ESRD patient studies, conference extracts, and case reports. Forty-nine studies were obtained for full references, and 27 studies were rejected because the population was mixed with other diseases and/or included a population other than the patient, such as family, health professionals, and caregivers.

Finally, 22 studies (See Table 2) were included in the appraisal and meta-synthesis process. A total of 532 ESRD patients were included in this study, with an age range of 30–92 years old, most of whom were elderly and late stage. The included studies were completed following CASP items, and most of the studies (18 studies (81%)) clearly described the rigor analysis process through member consensus or triangulation. Eight studies used NVivo software analysis, while one study used Atlas software analysis in the data analysis process. Approximately 81% of the studies included the number of male and female participants, consisting of 228 male participants and 179 female participants.

Regarding the method, all studies used qualitative methods, including semi-structured interviews [35,36,37,38,39,40,41,42], qualitative in-depth interviews [43,44,45,46], qualitative explorative and descriptive designs [47,48], and qualitative phenomenological hermeneutical designs [49,50,51]. Three studies consisted of ethnographic phenomenology studies [52,53,54], one study used grounded theory [38], and only one study used focus group discussion combined with personal interviews [55,56].

Exploring advance care planning was the main study purpose of the included studies [37,39,52,53], followed by exploring the treatment decision [38,41,42,43,46] and symptom management [35,36,44,47,50,51,54,56]. Two studies explored palliative care needs [35,49], and three studies investigated the end-of-life care perspective [40,45,55].

Seven studies explored patient experiences in the United States (US) and United Kingdom (UK), two studies in Sweden, one study in Canada, New Zealand, Malawi, Thailand, Malaysia, and one study explored Chinese patients in Singapore. Eighteen studies gathered data from renal centres in hospitals [35,36,37,39,40,41,42,43,44,45,46,47,48,49,50,51,54,56], two studies obtained data from clinics [52,53], and the other two studies analysed data from community centres [38,55]. The majority of participants were white (45%), followed by Latinos (17%), Caucasians (11%), and African Americans (8.5%). The other participants were Swedish, Asian, Malawi, Hispanic, Chinese, Black Caribbean, Black African, Aborigine, Mexican, and others.

**Table 2 ijerph-18-10651-t002:** Study Samples.

No	Sample	Study Purpose	Method, Sampling & Data Collection Technique	Sample Size (Male; Female; Age (Yrs))	Data Analysis (Technique; Rigor)
1	Al-Arabi, 2006 [50]	To describe how persons with ESRD experience and manage the quality of their daily lives.	- Naturalistic inquiry methods - Purposive sampling - Semi-structured interview	80	- Content analysis and constant comparative analysis - Consensus, Member check, Codebooks and Field Notes
2	Axelsson et al., 2012 [35]	To describe and elucidate the meaning of being severely ill living with haemodialysis when nearing end of life.	- Serial Qualitative Interview - Purposive Sampling - Serial interviews, open and clarifying questions	8 (5; 3; 66–87)	- 3 phases (naïve understanding, structural analysis, comprehensive interpretations) - Not described
3	Bates et al., 2017 [48]	To describe the palliative care needs of patients with end-stage kidney disease who were not receiving haemodialysis.	- Qualitative, explorative, and descriptive design - Purposive sampling - Semi-structured interview	10 (3; 7)	- Thematic analysis - Consensus
4	Beng et al., 2019 [36]	To explore the experiences of suffering of ESRF patients on maintenance dialysis in Malaysia.	- Qualitative study - Convenience sampling - Semi-structured interview	19 (15; 4; 30–60)	- Thematic analysis using NVivo9 - Consensus
5	Bristowe et al., 2015 [37]	To explore the experiences of people with ESKD regarding starting haemodialysis, its impact to quality of life and their preferences for future care and to explore the advance care plan needs of this population and the timing of this support.	- Semi-structured qualitative - Interview - Purposive sampling - Semi-structured interview	20 (11; 9; 62)	- Thematic analysis using NVivo software - Investigator triangulation - Clarification theme with participant
6	Bristowe et al., 2019 [47]	To explore the experience, impact, and understanding of conservatively managed end-stage kidney disease.	- Secondary analysis of qualitative interview - Purposive sampling - Structured interview	20 (11; 9; 82)	- Thematic analysis using NVivo software - Investigator triangulation - Expert review for the theme
7	Calvin, 2004 [38]	To explore decisions about end of life treatment in people with kidney failure undergoing haemodialysis.	- Grounded theory - Theoretical sampling - Open and Clarifying questions	20 (11; 9; 56)	- Constant comparative analysis - Memos - Member checking
8	Cervantes et al., 2017 [39]	To explore the preference of Latino patients receiving haemodialysis regarding symptom management and advance care planning.	- Qualitative study using semi structured interview - Purposive sampling	20 (10; 10; 61)	- Thematic Analysis using Atlas software - Consensus
9	Chiaranai, 2016 [51]	To understand the daily life experiences of Thai patients with ESRD who are on HD.	- Descriptive Phenomenological Study - Purposive sampling - Semi-structured interview	26 (8; 18; 48–77)	- Thematic Analysis - Consensus - Member checking
10	Davison, 2006 [52]	To determine perspectives of patients with end-stage renal disease of salient elements of advance care planning decision.	- Ethnographic, qualitative in depth interview - Purposive sampling - Semi-structured interview	24 (12; 12; 64)	- Constant comparative and iterative analysis - Not described
11	Davison & Simpson, 2006 [53]	To understand hope in the context of advance care planning from the perspective of patients with ESRD.	- Ethnographic, qualitative in depth interview - Purposive sampling - Open-ended questions	19 (11; 8; 64)	- Inductive analysis - Not described
12	Gonzalez et al., 2017 [55]	To explore intensive procedure preferences at the end of life in older adults.	- Convenience sampling - Semi-structured interview - FGD for non-ESRD patient	26 (14; 12; 70.6)	- Content Analysis - Member checking
13	Ladin et al., 2018 [40]	To examine how health literacy may affect engagement, comprehension, and satisfaction with end-of-life conversations among older dialysis patients.	- Qualitative descriptive study - Purposive sampling - Semi-structured interview - Open-ended questions	31 (15; 16)	- Team based consensus process using NVivo software
14	Lovell et al., 2017 [43]	Examined the experiences of older adults (aged ≥65 years) living with chronic kidney disease (CKD) as they chose whether or not to begin dialysis or continue with conservative management.	- Serial qualitative interviews - Convenience sampling - Semi-structured interview	17 (14; 3; 66–90)	- Thematic analysis using NVivo10 - Consensus
15	Petersson & Lennerling, 2017 [49]	To explore adults’ experiences of living with APD.	- Qualitative phenomenological hermeneutical - Purposive sampling - Open-ended interview	10 (82)	- Phenomenological hermeneutical method - Consensus
16	Russ et al., 2005 [54]	To explore lives and experiences of a number of individuals 70 years of age and older.	- Ethnographic phenomenology - Purposive sampling - Several times interview with semi-structured interview	43 (70–93)	- Thematic analysis - Not described
17	Seah et al., 2015 [41]	To gain insight into decision-making processes leading to opting out of dialysis.	- Qualitative study - Purposive sampling - Semi-structured interview	9 (84)	- Interpretative phenomenological analysis - Notes, Consensus
18	Sein et al., 2020 [44]	To explore patients’ experience of mild-to-moderate distress in ESKD.	- In-depth qualitative interviews - Purposive sampling - Semi-structured interview	46 (28; 18; <50–70)	- Thematic inductive analysis using NVivo - Consensus
19	Selman et al., 2019 [42]	To explore views and experiences of communication, information provision, and treatment decision making among older patients receiving conservative care.	- In-depth qualitative interview - Purposive sampling - Semi-structured interview	20 (11; 9; 82 (65–95)	- Thematic inductive analysis using NVivo - Consensus
20	Sharma et al., 2019 [56]	To uncover the lived experiences of this group of patients on centre-based haemodialysis (HD), the most prevalent dialysis modality.	- Qualitative Focus Group Discussion - Purposive sampling - Semi-structured interview	24 (14; 10; 57.4 ± 8.9)	- Thematic Analysis using Nvivo10 - consensus - inter-rater reliability using Cohen Kappa with K Value (0.80)
21	Sutherland et al., 2021 [45]	To explore the impact of the death of a patient in the haemodialysis unit on fellow patients.	- Qualitative Study - Purposive sampling - Semi-structured interview	10 (4; 6; 42–88)	- Thematic analysis - Consensus
22	Tonkin et al., 2015 [46]	Explore the experiences of older adults who had made a decision between different treatments for CKD stage 5 in 9 UK renal units.	- Qualitative Study - Purposive sampling - Exploratory Semi-structured interview	42 (28; 14; 82 (74–92)	- Thematic analysis using NVivo9 - Consensus - Field note

Five themes were identified from the included studies: struggling to face the disease, experiencing deterioration, overcoming the challenges of dialysis, leading to a positive outlook, and preparing for the end of life. In addition, fifteen sub-themes were identified from the included studies. Table 3 provides sub-themes and illustrative quotations from cited studies to clearly lay out the theme and sub-themes for the readers. Five themes are framed in Figure 2 that were interlinked with the patient’s experiences and feelings in living with dialysis near the end of life. The phase began with the experience in accepting the disease, which included struggling to face the disease followed by experiencing deterioration. The other phase started when the patients needed to deal with the treatment by overcoming the challenges of dialysis. During these experiences, some patients understood the meaning of the disease, which lead to a positive outlook. Finally, by understanding their condition, the patients tried to prepare for the end of life. 

### 3.1. Struggling to Face the Disease

Struggling to face the disease means exerting more effort in accepting the disease. Dialysis was a life-saving procedure for these patients, although some of the patients had a different opinion.

#### 3.1.1. Negative Feelings Associated with the Disease

Several negative feelings about the meaning of the disease included feeling trapped, denial, fear, grief, and being punished by God.

#### 3.1.2. Dialysis to Stay Alive

Facing the disease improved the patients’ knowledge about their condition. Dialysis had been viewed as dependence or addiction treatment to maintain life. However, some patients viewed dialysis as damaging to the body.

### 3.2. Experience Deterioration

The deterioration occurred due to the disease, which caused difficulties in maintaining quality of life since several changes occurred. The changes meant new challenges to face.

#### 3.2.1. Changes in Functional Status

Several physical changes occurred that affected respiration, fluid and electrolyte balance, and nutrition. Some symptoms that the patients experienced included fatigue, breathlessness, pain, insomnia, nausea, and vomiting.

#### 3.2.2. Changes in Emotional Status

Sensitivity increased when dealing with the disease and its effect on quality of life. Patients tended to experience sorrow and depression.

#### 3.2.3. Changes in Social Status

Social relationships with family, friends, and work changed dramatically due to decreases in body function and emotional changes. Some patients might lose their job, lack time with friends, lose their relationship with their sexual partner, and be dependent on their family.

### 3.3. Overcoming The Challenges of Dialysis

Burden in the process of treatment, the needs during treatment, and the effect of treatment on quality of life were major challenges that patients faced when undergoing life-saving dialysis. These challenges are continuous since the treatment continues until the end of life. Overcoming challenges was needed to continue the treatment and maintain the patients’ life.

#### 3.3.1. Facing Difficulties in Treatment Decisions

The different views indicate dialysis as a natural or invasive procedure affecting patient treatment choices. Autonomy was an important part of treatment decisions. In making decisions about treatment, there were some difficulties faced by patients and their families.

#### 3.3.2. Lack of Control of Daily Life

The changes due to the effect of the disease also affected the patients’ daily life, especially during the caring process, which is time consuming and limits their activities.

#### 3.3.3. Financial Strains

Financial strains resulted from the treatment cost for transportation, drugs, and the treatment fee itself.

### 3.4. Leading to a Positive Outlook

The process of dealing with the disease finally leads to a better outlook to find a positive meaning for the disease. Improvement in spiritual value and relationships, accepting the present situation, and preparing to make the best efforts for their future improves the patients’ life and results in a positive outlook for the disease.

#### 3.4.1. Increasing Attachment to God

An increased attachment to God was believed to maintain the patients’ hope in improving their health and condition.

#### 3.4.2. Building Relationships with Healthcare Professionals

Healthcare professionals were an important part of the caregiving process. Since the person-centred care has been implemented, the dialysis process and communication with the healthcare professionals enhance the relationship and build the patients’ trust to share their feelings and experiences about the disease.

#### 3.4.3. Accepting the Present Quality of Life

Accepting all the changes and dealing with the recent experiences will lead to better preparation for the next life.

#### 3.4.4. Predicting the Future

Knowledge about the disease prepares patients about what to expect in their future since their future is otherwise uncertain.

### 3.5. Preparing for the End of Life

Acceptance of the disease was the key to preparing end-of-life care through further information about their life expectancy, awareness of mortality, and discussions about end-of-life issues, such as advance directive planning.

#### 3.5.1. Information about Prognosis

Early and more information about the disease and its development are helpful for patients to have informed expectations of their future.

#### 3.5.2. Awareness of Mortality

By understanding the disease and prognosis, patients are aware of the life-limiting nature of the disease.

#### 3.5.3. Talking about ACP Issues

Advance care planning (ACP) is a common issue discussed during preparation for end-of-life care. The ACP discussion involved the whole process of the patients’ care, including family and healthcare professionals.

## 4. Discussion

In facing a chronic disease with life-limiting potential, several studies mentioned that patients experienced a stage of grief and fear that required rapid adjustment [37,39,54]. A study mentioned the sub-theme fatalism: the sense that one’s illness deserved punishment, suggesting that the disease was God’s punishment [39]. Depression might occur in the young population, most of whom chose to undergo dialysis or transplantation [57,58]. Elderly patients with limitations preferred to avoid further damage from dialysis by choosing conservative management [19]. Struggling with the disease resulted in gaining more knowledge about the disease [49]. In terms of accepting that dialysis was needed to stay alive, there were different opinions reported by the studies. Several sub-themes showed the patients’ perspective on their acceptance of life on dialysis [49,52,54]. In contrast, there were several sub-themes in choosing life support, showing that some patients did not want to spend their life being dependent on a dialysis machine. Sub-themes mentioned the patients’ fear of being addicted and dependent on dialysis for the rest of their life [41,45,50,51,54].

Deterioration affected the physical, emotional, and social status changes. Most studies showed that changes in functional status included pain and insomnia, breathlessness, nausea and vomiting, and fatigue. Two studies defined fatigue as an experience of having no energy and strength, and the patients were struggling to deal with the symptoms. Emotional status changes in accepting the disease may lead to a delay in or withdrawal from treatment [35,36,44,48,50,51]. In performing dialysis, studies have found that patients will lose their personal relationships, such as the relationship with their role as sexual partner, social relationships with friends and family, such as careers and breadwinners, and relations at work as an important part of companies. These changes were affected by time limitations and their physical condition itself. Feelings of dependency on caregivers or families has also developed due to the increasing schedule for visiting healthcare professionals to perform treatment while they are unable to travel by themselves [36,37,47,48,50,51].

Studies mentioned the contradiction of seeing dialysis as a natural vs. invasive procedure before making a treatment decision [46,55,56]. Patients have autonomy in making their decision. A study identified a theme with “immediate rejection of dialysis” in showing the decision-making process [41]. The complexity of the treatment, including the treatment process, treatment effect, and adherence to food and fluid restrictions, were the biggest challenges faced during dialysis. Somehow, this image was a scary thing for the patients that made them feel threatened and decide not to undergo dialysis [2,50,56]. Conservative therapy is seen as a less-aggressive therapy to support the patients’ life [59]. Due to dialysis, patients will lose their economic spheres due to a lack of time to be productive [39,40,51]. Other financial difficulties also grew from the need to afford modified food for diet and medications [48].

Several studies in which most patients accepted changes in their lives started to worry about uncertainty in the future [37,43,47,48,49,50,51,56]. There was an increase in building relationships with health care professionals. Participants put their trust, built more communication, and felt satisfied to obtain person-centred care, which improved their health from nurses who were most of the time in the treatment process [37,39,44,45,49,53,55]. Clear explanations about modality treatment from doctors gain the patients’ knowledge about the available options to keep their life [60]. Together with their positive development, the theme importance of spiritual and cultural beliefs shows the patient’s hope through prayer [48,50].

Comorbidities and survival rates in dialysis shorten the life expectancy of people without renal transplantation [11]. This review showed that studies mention the patients’ awareness about their own mortality, and they were aware that their position was between life and death [37,38,45,54]. Pre-dialysis education will help to slow or stop progression. The information needed by the patients was information about their disease and prognosis and several information preferences [40,42,52]. This information will then lead to advance care planning discussions within patients, families, and healthcare professionals to prepare their end of life [39,52,53,55].

### 4.1. Implications for Practice

The findings can be considered important for nurses and nursing care to improve palliative care for ESRD patients. ESRD patients have to struggle more than those with other diseases, where only limited options are available to help them live longer. At first, after being diagnosed, most patients tried to deal with their own disease and their own thinking to find the reason why they got the disease. In this situation, nurses can explain more about the disease, such as the pathophysiology of the disease and especially the potential causes of the disease to help them understand about the disease and decrease their negative feelings. Also, nurses can provide information about the possibility of treatment choices and their availability and each treatment’s advantages and disadvantages.

Along the disease development, patients might experience changes in functional, emotional, and social status which will impact on their quality of life. Nurses need to care for the symptoms based on the patient’s need since every patient might experience different functional changes according to the disease development. In building positive emotion, nurses should determine patients’ emotional status, build trust, avoid sensitive things in communication, and encourage them to have better feelings of their self in dealing with the disease. Since nurses are an important part of treatment, nurses also need to help patients in re-building their social life by giving social support and discussing potential solutions to adjust their social life changes.

Patients have their own autonomy in choosing their treatment options. Some patients might refuse to undergo dialysis, preferring instead to choose conservative management, although most of them choose dialysis to keep alive. Nurses should respect the patients’ and families’ decision and continue to help them with the consequences of it. Comprehensive information about the treatment, possible consequences of the treatment, and how to handle the consequences might lessen patients’ fear about the treatment. Sharing experience from other patients from each treatment chosen might be a benefit for other patients to support their own treatment decision.

At the acceptance stage, where patients have accepted their situation, nurses can persuade them to reward themselves of their efforts in managing themselves in combating the disease. Spiritual support also can be provided regarding patients’ need. Therapeutic communication should be maintained to keep a good relationship in order to prepare future plan care since patients are aware about the uncertainty of their life. 

Important information in preparing the end-of-life care was information about the disease progress. Patients’ awareness about living between life and death will be helpful to lead communication about the advance care planning. Nurses can be a bridge of communication between patient, family, and healthcare professionals to build advance care planning.

This study highlighted the value of patients’ views for informing clinical practice which can be a benefit to know about patients’ perceptions regarding the disease, how they make decisions on treatment while thinking about their uncertain future, and death. End-stage condition puts the patients’ needs for more informational support from nurses about their future condition that might increase patients’ engagement during the palliative care process.

### 4.2. Strength and Limitations 

A quality appraisal has been applied in this study to check each study rigor using CASP, which addresses all the principles and assumptions underpinning qualitative research [31]. Those included studies represent for various countries and various qualitative study methods which support for transferability or generalizability of evolving themes. RAMESES checklist has been used to ensure comprehensive reporting [27]. However, patients with ESRD have complex experiences. Since this study was a secondary data analysis study that depended on other research analyses, the researchers did not have the opportunity to have direct personal communication to explore more information and insight into the feelings of each participant. Most of the studies were conducted in Western countries and more than half of the population was white. Palliative care for ESRD needs may vary among cultures, and further studies with minority populations are recommended.

## 5. Conclusions

This study focused on the patient’s experience which can be used as a reference for palliative care for ESRD patients. The results showed that informational support from nurses was an important consideration for patients to deal with the diagnosis, choose the treatment, and plan for their care. To provide better care, it is suggested that detailed information about the disease and prognosis should be provided by nurses at hospital. Results from this study also can be used to develop interventions to improve the quality of nursing care for ESRD patients, such as developing an educational package for preparing for end of life, including information about disease prognosis, building awareness of mortality, and discussions about ACP issues by involving patients and their families. Developing a standard care at hospital will lead a better way for nurses to provide care. In addition, developing a curriculum based on the patient’s needs will be important to prepare knowledge in nursing higher education. Future synthesis studies may consider literature related to culture and religion in implementing end-of-life care for end-stage renal disease patients and/or families from nursing disciplines and other studies.

## Figures and Tables

**Figure 1 ijerph-18-10651-f001:**
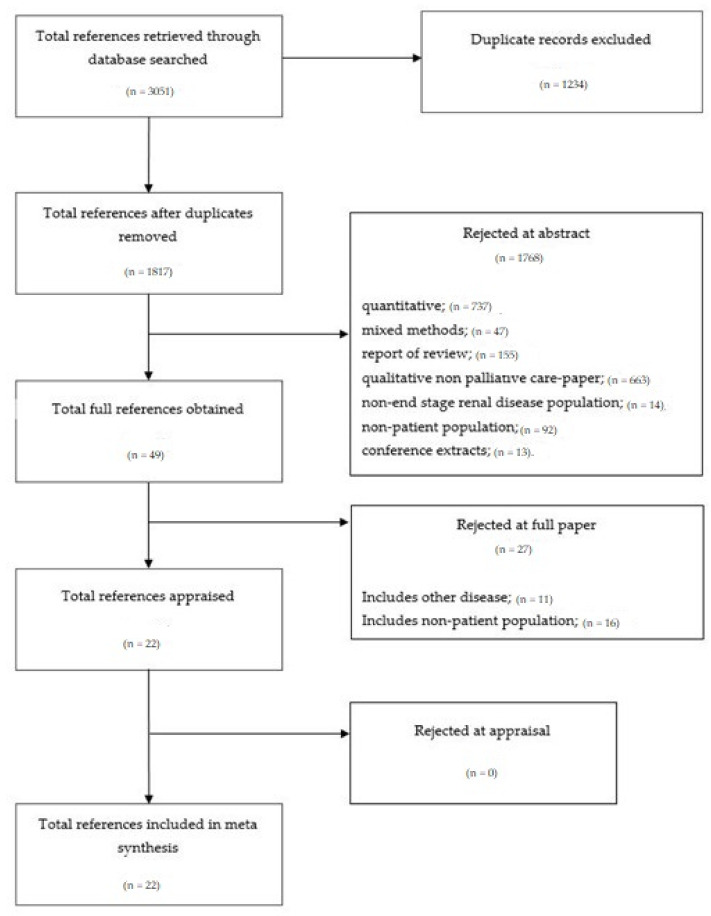
Flowchart of the meta synthesis steps.

**Figure 2 ijerph-18-10651-f002:**
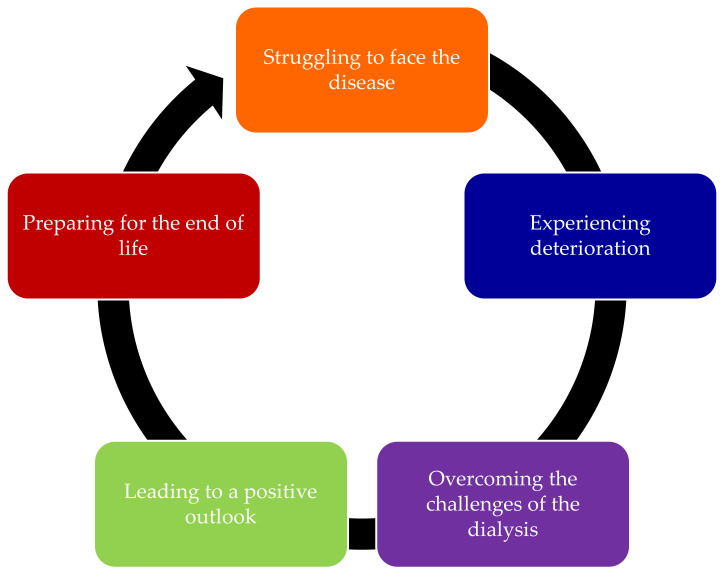
Framework of the experiences of patients with end-stage renal disease near the end of life.

**Table 1 ijerph-18-10651-t001:** Algorithm used in the PubMed Web search.

‘kidney failure’ OR ‘renal failure’ OR ‘chronic kidney’ OR ‘chronic renal’ OR ‘chronic nephropathy’ OR ‘chronic renal disease’ OR ‘CRD’ OR ‘late CRD’ OR ‘late stage CRD’ OR ‘late stage chronic renal disease’ OR ‘terminal CRD’ OR ‘endstage renal disease’ OR ‘end-stage renal disease’ OR ‘ESRD’ OR ‘chronic kidney disease’ OR ‘CKD’ OR ‘late CKD’ OR ‘late stage CKD’ OR ‘late stage chronic kidney disease’ OR ‘terminal CKD’ OR ‘endstage kidney’ OR ‘end-stage kidney disease’ OR ‘endstage kidney disease’ OR ‘ESKD’
AND
‘palliative care’ OR ‘End of Life Care’ OR ‘Hospice Care’ OR ‘Supportive Care’ OR ‘Conservative Care’ OR ‘Non-dialysis Care’
AND
‘qualitative’ OR ‘life experience’ OR ‘narratives’ OR ‘interview’

**Table 3 ijerph-18-10651-t003:** Illustrative Quotations by Theme.

Sub-Theme	Citation	Representative Quotations
**I. Struggling to face the Disease**		
Negative feelings associated with the disease	- Feeling trapped [35] - Acceptance of the condition and future [48] - Looking back: emotions of commencing HD [37] - Beating the odds [38] - Fatalism: the sense that one’s illness is deserved punishment [39] - Sense of Personal Empowerment [52]	‘What can you expect? That something happens quickly? Will it be slow and squeeze the life out of you or will it be fast? That’s what one’s thinking about’ ‘It doesn’t end, this, because it will go on and on and on, and I will never get well, I will never escape the dialysis... the night before I go [to dialysis treatment]... I think, what’s the point really of me going?’ [35] I just didn’t really want to live anymore because I thought I can’t live a life like this. It was so difficult in the beginning … you wouldn’t imagine how difficult it was. [37]
Dialysis to stay alive	- Staying Alive [50] - Feeling disconnected from it [47] - Improved understanding of illness leads to adherence [39] - Feeling safe while undergoing HD treatment [51] - Discordant Expectations and Dialysis Experiences [40] - Needing dialysis in order to survive [49] - ‘Choosing’ Life (Support) [54] - Personal ownership of decision [41] - Acceptance of dialysis [45]	“Actually, I’m not obligated (to have dialysis)... I want the symptoms to improve, to be better, because if I don’t come (to dialysis), I know I’m going to feel worse. I’m going to feel worse because the liquid stuff affects me” [39] “Without dialysis I would not have survived until now.” [49] ‘There is life on dialysis,’ ‘you are willing to change yourself and your expectations. IF you are willing to recognize yourself as you are rather than as you hoped to be.’ [54] It’s a big commitment, but it keeps me alive. [45]
**II. Experience Deterioration**		
Changes in functional status	- Tied Down [50] - Interpreting the deteriorating body [35] - Physical Suffering [36] - A decrease in physical activity [51] - Information about the Impact of Interventions on Daily - Life [52]	‘When I walk a small distance I have to rest because of breathlessness.’ ‘I feel nausea all the time and sometimes I vomit.’ [48] “With my kidneys… it’s, well…hard to say, I just feel tired and no energy.” My knees and hips were so weak that I didn’t dare climb the stairs any more. If I fall, you know, I would hardly be able to get up again. [49]
Changes in emotional status	- Depression [48] - Psychological Suffering [36] - Struggling to attribute symptoms to the illness [47] - Dealing with emotional change such as anger, guilt, depression, and unhappiness [51] - Coping techniques [41] - The emotional burden of distress [44] - Feelings about the decision [42]	‘I was very devastated the time I was told that my kidneys are damaged and will not work normally.’ [48] They put a line in me… cause I had to get on the dialysis straight away, then they had the ER about doing the bags. Oh, I cried my eyes out, I was terrified when all this at the beginning was going on. I was petrified. [37] “Mentally, I was very depressed when they told me I couldn’t have the fistula… that was ten months ago, so I was very depressed then, but I didn’t agree with it, certainly it wasn’t right what they said about the two months to go… but now, I don’t know how far I’m going to go.” [42]
Changes in social status	- Left Out [50] - Having a changing social life [35] - Loss of role within the family [48] - Social Suffering [36] - Impact on friends and family [37] - A narrowed social life [51] - Enhancing relationships [53] - Patients Conforming to Social Roles [40]	You know they can’t get on with their life cause I can’t get on with mine, cause I’m stuck on this. Too busy helping me out with my little girl. So it’s a lot of strain and pressure yes on the family and friends. [37] ‘This disease has really affected my family life because I do not have sexual feelings these days. I don’t have strength, I feel weak and breathless.’ ‘I used to do farming but now I cannot because I do not have strength.’ [48] “That was the worst thing with my company, when I got sick the phone calls stopped…you’ve worked over 20 years for a company and all of a sudden you’re like a piece of paper in the wind” [53]
**III. Overcoming the challenges from the dialysis**		
Facing difficulties in treatment decision	- Involvement in treatment decisions [37] - Flexible decision-making conversations at home with family [39] - The acceptability of a “natural” versus “invasive” procedure [55]	‘Well, the doctor did encourage me to go on dialysis, but I said “no” I knew from the start that I did not want to go through dialysis if my kidneys ever failed’ ‘Then I discussed it with my family… My life belongs to me, but also to everyone else, so I discussed with them…’ [41]
Facing difficulties in treatment decision	- Autonomy [49] - Balancing odds and reaching decision: age and life completion - financial and physical - burden of dialysis [41] - Reasons for treatment decision [42] - Treatment imposition [56] - Patients’ experiences of making a management - decision about their CKD [46]	It’s a big risk that we’re going through, in my opinion, being on the machine. Anything can happen because they’re messing with our lives, you know. They are doing the best they can do, [for] which thank God. But still, it’s a risk that we’re taking… It just worries me. [38]
Lack of control of daily life	- Doing Without [50] - Losing control in life with illness [35] - Impact on day-to-day life [37] - Discovering meaning [38] - Dietary restriction is culturally isolating and challenging for families [39] - Discordant Expectations and Dialysis Experiences [40] - ‘Doing Time’’ For Dialysis, For Life [54] - Treatment imposition [56]	Everything’s changed, every single thing … Well I can’t walk, I can’t eat everything what I fancy, I can’t drink really what I want … to drink. Oh life stinks, horrible, can’t stand it. Terrible times this is. Doesn’t hurt having it done … but oh my god Sunday nights, they’re a git. [37] “One of the hardest things about dialysis is the diet. I don’t eat bananas, I don’t eat oranges” “At the beginning, everything was off. I was getting cramps… I feel better now. That first year was awful” [39] ‘…I visited my friend, who’s a kidney patient, after the dialysis… water dialysis, they put the tube in and take out, then she’s got diabetes blister, she suffered a lot. Whole life. No changing. Nothing. I said I’d better die, whole life no cure, no nothing, no point, I say…’ I’m very happy! Going on dialysis is like staying in prison’ [41]
Financial Strains	- Financial challenges impacting hospital care [48] - Logistic challenges and socioeconomic disadvantage compounded by health literacy and language barriers [39] - Spend hidden cost related to HD treatment [51] - Physical and financial burden of condition [41]	‘It is very difficult for me to find money for transport to come to the hospital because I came with my guardian and we use MK3700.00 (approximately $8) per visit which is also difficult because now I cannot work.’ ‘Every time when you go to get some medication in our government pharmacy, they [medications prescribed] are not available. They [health workers] tell us to go and buy. The medication is very expensive [$5] which I cannot afford. I just go back home without the medication.’ [48] “Sometimes I come here with not enough food because of the financial problems that I have” [39]
**IV. Leading a positive outlook**		
Increased attachment to God	- Trust in God [50] - Hope [48]	‘Prayer is powerful because in whatever I have been through … I know with God everything is possible. With the problems I have been through they make to be closer to him.’ [48] “We just have to trust in the Lord because he, and it’s up to God to, ah, heal us. Just like it may be some miraculous healing that he can bring to a person. Because anything is possible with God. There’s an impossible with man. All things are possible with God. So I believe this.” [50]
Building relationships with healthcare professionals	- Sources of support [39] - Empathetic listening and affirming self-worth of patients [52] - Partnership in care [49] - Patient-staff interactions and kidney unit support [44] - Participant views and experiences of staff-patient communication [42] - Nurses and the haemodialysis community [45]	“I just put my trust in the doctors. I trust that they know what they are doing.” [52] “I mean it’s, it’s a very, very nice feeling to be able to go there and know… that you can ask them any question you like. It doesn’t matter how it sounds to be but to you it’s important and they answer you and they’ve got time, that’s the thing, time for you…” “(The dialysis team) reassured me, told me not to be afraid; they said that there are people who’ve been on dialysis for over 20 years and they’re still here, so that encouraged me” [39]
Accepting the present quality of life	- Accept it as part of life [50] - -Having to accept a changed life [35] - Realisation [37] - Focused on daily life [53] - Quality of life [40] - Focusing on life [49] - Coping strategies [56]	You tend to be in a state of denial … We have to handle ourselves and say, right, we have to do this. There’s going to be days where we don’t want to do it. We’re going to overcome this. We have to really get to realise, this is what’s keeping us alive. [37] There is still much to enjoy in life. When one sees how difficult it is for many others. I think that life is worth living! As long as I can read a book, do crossword puzzles, and be social with friends. [49]
Predicting the future	- Talking about future care [37] - Feeling insecure that HD treatment will not last for long [51] - Hope shapes both goals of care and advance care planning [53] - Clinical indicators of kidney function [43] - Uncertainty about the future [49]	It’s something you accept in the end. The insight that you have an unknown number of years ahead of you. It may not be that many, or perhaps a few more. But this gives you a different view of … what life is about. [49] So you, you do your best for a while, and I feel okay, so it’s alright. See, kidney failure is a thing that just creeps up gradually so you just, you cope with it day by day, not realizing that you’re getting worse and worse all the time. So I know other people of my age that, I’m not functioning like they are. They’re having a good time. I can’t mow my lawns and all that sort of stuff, so that’s why I’m here [training for dialysis]. [43]
**V. Preparing for end of life**		
Information about prognosis	- More Information about Prognosis and the Disease Process Earlier in the Illness [52] - More information [53] - Prognosis [40] - Gaining necessary knowledge [49] - Information preferences [42]	“I would hope that healthcare providers are sufficiently trained to inform the patients at the right time what to expect and not wait until the very last minute” “They never told me how long I’d live…on dialysis. How long that will be?” [40] One of the things I’d like to know is how long my kidneys will last but they can’t tell you that exactly, and they said four or five years, so that’s not their fault is it? I want to sit down and get real answers, but the answers aren’t always there, are there?” [42]
Awareness of mortality	- Facing own mortality [37] - Knowing the odds [38] - Individualised [53] - Level of trust in physicians and autonomy in decision-making [55] - Reconciling EOL values and plans for future care [40] - Acceptance of death & Patients preparing for the eventuality of death [45]	‘It’s just a matter of time’, ‘I’m not gonna [going to] live forever’ and ‘We all have to do it’. Patients also described their postmortem preparations (i.e., funerals, cremations, burials, caskets, obituaries and cemeteries) [38] “Mortality is only one phase of our existence... the end of life is no more traumatic that the beginning of life... your only concern is leaving loved ones here for a brief time. My concern is for my wife but… it is not a frightening factor” [53] “I already have a burial plot at the cemetery, near my husband, I have money for each of my children, I have asked my children to bring the mariachi, and the songs I wish for them to play, and I told them how to divide the land I have in Mexico. I’m ready, and this way, I avoid discord between my children.” [55]
Talking about ACP issues	- ACP conversations incorporating trust and linguistic congruency [39] - Patient’s Perceived Benefit of ACP [52]	“My wife doesn’t want to touch the subject (of ACP), but I think it does help me because one is worried about the family and what’s going to happen and this and that. My wife tells me, ‘no, don’t worry’” “Talking about [ACP] lets you know what’s going to happen. I need to know what the symptoms are and he wouldn’t tell me… because I’m really worried about nausea, vomiting, and not being able to breath. Someone should be talking to you about what’s coming.” [52]

## Data Availability

Not applicable.

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
