# Peer review of "Palliative Care in Patients with End-Stage Renal Disease: A Meta Synthesis"

_ijerph, 2021, doi:10.3390/ijerph182010651_

Round 1
Reviewer 1 Report
Dear authors,
Some aspects should be addressed to enhance readability and understandability:
- the given studies being included should be pooled in 1 synoptic table, instead of being presented on 7 consecutive pages (page 6-12), which is somehow a little bit confusing;
- in the conclusion section you state that results from this study can be used to develop interventions to improve quality of nursing care for ESRD patients: some precise examples should be given
-the baseline characteristics of included patients should be shown more precisely in a table to illustrate the patients cohort to the reader, so that the reader can compare the included patients with his/her own ESRD patients population
Author Response
Dear Editor,
Many thanks for your interest in our manuscript entitled “Palliative Care in Patient with End Stage Renal Disease: A Meta Synthesis”. We wish to thank you for your suggestions. We are extremely grateful to you and the reviewers for your valuable comments, which have enabled us to make substantial improvements to the paper.
A revision of this manuscript has been made and enclosed with this letter. To make it easier for you to follow how the changes were made. In addition, a table has been included that specifies each comment made by the reviewers.
We do hope that the quality of this version is much improved. Your quick response of the status of this manuscript would be highly appreciated. We are looking forward to hearing from you soon. Your kind efforts in helping international scholars to promote the quality of international academic societies are highly appreciated.
Sincerely,
Authors

Reviewer 2 Report
This is a very interesting approach to end-of-life care in renal patients depending on dialysis. The main suggestions I have are related to table 2. Reference 4 (Beng et al): what is the relevance to describe in “Sample Size & Key Characteristic” the religion of patients? It could be useful if all studies have the individual religions, which allow us to assess the religious influence on end-of-life decisions. The same goes for the language, reference 17 (Seah et al). My suggestion is that beyond demographic data, the authors should include common or equivalent variables to reduce interpretation bias.
Table 3, although extensive, reveals the patients's feelings, being very important for the qualitative analysis facing a terminal disease.
Author Response

(The authors gave the same response as above.)

Reviewer 3 Report
The qualitative meta-synthesis presented by Nur Fithriyanti Imamah et al. primarily focuses on palliative care in End-Stage Renal Disease (ESRD) patients. The authors provide valuable insight into the experience of ESRD patients, identifying five themes and fifteen sub-themes from 22 studies included in the meta-synthesis. The authors' interpretive integration of qualitative findings could be used as a reference to improve the quality of the palliative care received by ESRD patients. Overall, the article is interesting and well-written. Here are some comments to improve the manuscript:
As a strength of their study, the authors mention that the RAMESES checklist has been used to ensure comprehensive reporting (Line 383). However, this information is not included in the methods section.
In the results section, the authors describe the five themes identified (struggling to face the disease, experiencing deterioration, overcoming the challenges of dialysis, leading to a positive outlook, and preparing for the end-of-life). However, the integration of their findings does not reference the specific studies from which these observations were derived. The inclusion of these references would provide clarity and support their results.
At the end of the discussion, the authors include a section of the strengths and limitations of their study. Is the complexity of the ESRD experience a limitation of their study? Please elaborate
Minor comments:
Abbreviate terms when first mentioned and be consistent throughout the manuscript (particularly with the abbreviation for ESRD)
-Line 41: Murtagh et al. is not properly referenced
-Lines 61-62: This line lacks references
-Line 105: The authors mention that databases were searched individually using both search terms. However, they describe three cluster keywords. Please correct.
-When software is mentioned (EndNote, Nvivo, Atlas), please include information regarding the developer and the software version when available.
-Line 174: remove the before Sweden
-Line 354: remove of
-Line 358: possibility -> posible
-Lines 359-361: unclear paragraph
Author Response

(The authors gave the same response as above.)

Round 2
Reviewer 1 Report
Dear authors,
you satisfactorily addessed the remarks/comments.